# Association between the proportion of laparoscopic approaches for digestive surgeries and the incidence of consequent surgical site infections, 2009–2019: A retrospective observational study based on national surveillance data in Japan

**Toshiki Kajihara**[1]*, **Koji Yahara**[1], **Aki Hirabayashi**[1], **Yumiko Hosaka**[1], **Norikazu Kitamura**[1], **Motoyuki Sugai**[1], **Keigo Shibayama**[2]

**1** Antimicrobial Resistance Research Center, National Institute of Infectious Diseases, Tokyo, Japan,
**2** Department of Bacteriology/Drug Resistance and Pathogenesis, Nagoya University, Graduate School of Medicine, Showa-ku, Nagoya, Japan

* kajihara@niid.go.jp

## Abstract

### Background

Surgical site infections (SSIs) are among the most common healthcare-associated infections. Laparoscopy is increasingly being used in various surgical procedures. However, no study has examined the association between the proportion of laparoscopic procedures and the incidence of SSIs in digestive surgery using nationwide surveillance data.

### Methods

We retrospectively investigated national SSI surveillance data from the Japan Nosocomial Infections Surveillance between 2009 and 2019. The annual trend of the SSI rate and the proportion of laparoscopic procedures were assessed, focusing on five major digestive surgeries. This was based on data from 109,544 (appendix surgery), 206,459 (gallbladder surgery), 60,225 (small bowel surgery), 363,677 (colon surgery), and 134,695 (rectal surgery) procedures. The effect of a 10% increase in the proportion of laparoscopic procedures on the reduction of the SSI rate was estimated using mixed-effect logistic regression.

### Findings

The average SSI rate of the five digestive surgeries decreased from 11.8% in 2009 to 8.1% in 2019. The proportion of laparoscopic procedures in each of the five digestive surgeries increased continuously (p<0.001). The SSI rate for laparoscopic procedures was always lower than that for open procedures. The results were consistent between all and core hospitals participating in the surveillance. The odds ratios of the 10% increase in the proportion of laparoscopic procedures for five digestive surgeries were always <0.950 (*p*<0.001).

**Data Availability Statement:** All data to replicate the study's findings are available at https://github.com/bioprojects/laparoscopy_and_SSIs.

**Funding:** This study was supported by the Research Program on Emerging and Re-emerging Infectious Diseases from the Japan Agency for Medical Research and Development (AMED) under grant number JP21fk0108604. Funding acquisition: K. S. and M. S. The funders had no role in study design, data collection and analysis, decision to publish, or preparation of the manuscript.

**Competing interests:** The authors have declared that no competing interests exist.

**Abbreviations:** APPY, appendix surgery; BILI, Bile duct, liver or pancreatic surgery; BRST, breast surgery; CARD, cardiac surgery; CBGB, coronary artery bypass graft with both chest and donor site incision; CHOL, gallbladder surgery; COLO, colon surgery; CRAN, craniotomy; FX, open reduction of fracture; GAST, gastric surgery; HER, herniorrhaphy; HPRO, hip prosthesis; JANIS, Japan Nosocomial Infections Surveillance; KPRO, knee prosthesis; LAM, laminectomy; NEPH, kidney surgery; PRST, prostate surgery; REC, rectal surgery; SB, small bowel surgery; SSI, surgical site infection; THOR, thoracic surgery; XLAP, exploratory surgery.

## Conclusion

An increase in the proportion of laparoscopic procedures was associated with a reduction in the SSI rate in digestive surgeries.

## Introduction

Surgical site infections (SSIs) are among the most frequent healthcare-associated infections worldwide [1]. SSIs are also associated with increased morbidity, in-hospital mortality, and prolonged hospitalization. On a per-case basis, surgical site infections amount to $20,785 (95% CI: $18,902–$22,667) [2]. The SSI rate varies from 1% to >20% for each surgical procedure. The SSI rate of digestive surgeries is higher than that of other surgeries because most digestive surgeries include intestinal manipulation [3, 4]. The associated SSI rates are as high as 14–25% in colon surgery [3, 4].

Laparoscopic techniques have been increasingly used in digestive surgeries, leading to fewer postoperative complications. Laparoscopic procedures are minimally invasive. In a systematic review, Buia et al. showed that the advantage of the procedure was minimizing trauma to the abdominal wall compared with open surgery [5]. It was based on another systematic review that pain intensity on VAS scale was reduced on day one and wound infections were decreased in laparoscopic procedures for suspected appendicitis [6]. Laparoscopic procedures accelerated the recovery after colon surgery by decreasing pain and duration of hospital stay [7]. van der Pas et al. reported that patients in the laparoscopic colectomy group lost less blood, bowel function returned sooner, and hospital stay was shorter than those in the open surgery group in a randomised, phase 3 trial [8]. Laparoscopy has been shown to be associated with a lower risk of SSI [3, 4, 9]. The benefit of the laparoscopic technique compared to the open procedure with respect to SSI has been reported in digestive surgery, especially in colon surgery: Caroff et al. reported that the incidence of SSIs in laparoscopic and open procedures was 4.1% and 7.9%, respectively, in a cohort study including 229,726 cases [10]. Athanasiou et al. performed a systematic review and meta-analysis of appendectomy with complicated appendicitis; they found that laparoscopic procedures decreased the incidence of SSI compared to open procedures (odds ratio 0.30) [11]. Recently, Kayano et al. showed the usefulness of the laparoscopic technique in small bowel surgery: the SSI rate in laparoscopic procedures was 3.8% and that in open procedures was 26.9% in a retrospective study including 105 cases in Japan [12].

However, no study has examined how the proportion of laparoscopic procedures in each type of digestive surgery is associated with the reduction of the incidence of SSI using nationwide surveillance data. We utilized SSI surveillance data collected in a national surveillance program, the Japan Nosocomial Infections Surveillance (JANIS), involving hundreds of participating facilities. This retrospective study aimed to investigate how the spread of laparoscopic procedures is associated with the reduction of the incidence of SSI for five digestive surgeries, using Japanese national surveillance data from 2009 to 2019.

## Materials and methods

### Data preparation

The JANIS SSI section is one of the largest voluntary SSI surveillance systems [13]. It was established in 2002. The program was developed according to the principles of the National

Nosocomial Infections Surveillance (NNIS) system known as the National Healthcare Safety Network (NHSN). The SSI surveillance in JANIS is conducted using the definition of the US National Nosocomial Infections Surveillance System (NNIS), with some modifications. It is comparable to other international SSI surveillances, such as the OP-KISS of Germany [14] and the National Healthcare Safety Network of the United States of America [15]. Surgeons in hospitals performed procedures and diagnosed SSI cases and reported the data to JANIS according to NHSN Operative Procedure Categories using ICD-9-CM code (https://janis.mhlw.go.jp/section/master/surgicaltechniquecode_ver2.1_20121220.xls). More hospitals progressively joined the JANIS SSI section, and the program included 785 hospitals in 2019. These included 188 of 408 (46.1%) hospitals with over 500 beds, 464 of 2,174 (21.3%) hospitals with 200–499 beds and 133 of 5790 (2.3%) hospitals with less than 200 beds. Most patients were followed up at outpatient clinics in their hospitals in Japan. The surgeon is usually responsible for the course of procedures in Japan. This is because JANIS collects the data from hospitals. The surgeons and/or infection control nurses in the hospitals validated the data and submitted the data every six months. A total of 1,564,438 procedures were submitted to the JANIS SSI section between 2009 and 2019. The raw data were tabulated for each hospital participating in the surveillance for the following 18 surgeries using an in-house Perl script, and the aggregated data are available at https://github.com/bioprojects/laparoscopy_and_SSIs. Incidence rates of SSIs were calculated following 18 surgeries (appendix surgery (APPY), breast surgery, cardiac surgery, coronary artery bypass graft with both chest and donor site incision, gallbladder surgery (CHOL), colon surgery (COLO), craniotomy, open reduction of fracture, herniorraphy, hip prosthesis, knee prosthesis, laminectomy, kidney surgery, prostate surgery, rectal surgery (REC), small bowel surgery (SB), thoracic surgery and exploratory surgery). Bile duct, liver, or pancreatic surgery and gastric surgery were excluded because the aggregation method changed in 2012. Definition of SSIs including superficial incisional, deep incisional and organ/space SSI was based on the guideline of CDC's recommendations for prevention of SSIs [16].

We conducted analyses of data from either "all participating hospitals" or "core participating hospitals" (Table 1). The former included all hospitals participating in each year. In total, the numbers of participating hospitals for the five digestive surgeries were 395 (APPY), 424 (CHOL), 390 (SB), 628 (COLO) and 569 (REC) in 2019. Core participating hospitals included all hospitals consistently participating from 2009 to 2019. In total, the numbers of the core participating hospitals were 69 (APPY), 78 (CHOL), 72 (SB), 141 (COLO) and 120 (REC).

The SSI rates for all surgeries, for the five digestive surgeries, and for the other surgeries other than the five digestive surgeries for all and for the core participating hospitals were calculated for each year from 2009 to 2019. The proportion of laparoscopic procedures for the five digestive surgeries were calculated for each year from 2009 to 2019.

The proportion of SSIs for the five digestive laparoscopic and open surgeries were calculated from 2009 to 2019.

## Statistical analysis

Data were analyzed by Pearson's $\chi^2$ test or Fisher's exact test when the expected count in any category was >5, Continuous non-normally distributed variables were analyzed by the Mann–Whitney U-test. The Cochran-Armitage trend test was used to test whether there was a trend of the proportion of SSIs and laparoscopic surgeries across years. The effect of a 10% increase in the laparoscopic rate on the SSI rate was estimated using mixed-effect logistic regression analysis implemented in lme4 package [17]. The level of significance was set at $p < 0.05$.

**Table 1. Characteristics of five digestive surgeries.**

| | Appendix surgery (APPY) | Gallbladder surgery (CHOL) | Small bowel surgery (SB) | Colon surgery (COLO) | Rectal surgery (REC) |
|---|---|---|---|---|---|
| All participating hospitals | | | | | |
| No. of participating hospital in 2019 | 395 | 424 | 390 | 628 | 569 |
| No. of procedures between 2009 and 2019 | 109,544 | 206,459 | 60,225 | 363,677 | 134,695 |
| No. of SSI between 2009 and 2019 | 5,630 | 6,178 | 8,411 | 43,045 | 19,806 |
| No. of laparoscopic procedure between 2009 and 2019 | 66,246 | 165,921 | 9,068 | 160,864 | 72,701 |
| The rate of SSI of laparoscopic procedures, % | 4.19 | 1.91 | 8.00 | 7.27 | 11.3 |
| The rate of SSI of open procedures, % | 6.60 | 7.42 | 15.0 | 15.5 | 18.8 |
| Core participating hospitals | | | | | |
| No. of participating hospital | 69 | 78 | 72 | 141 | 120 |
| No. of procedures between 2009 and 2019 | 33,961 | 61,913 | 20,874 | 134,158 | 48,358 |
| Average no. of procedures of each year, n (±SD) | 3087 (±85) | 5628 (±485) | 1898 (±87) | 12196 (±1005) | 4396 (±241) |
| No. of SSI between 2009 and 2019 | 1,941 | 2,005 | 2,989 | 16,481 | 7,435 |
| No. of laparoscopic procedure between 2009 and 2019 | 17,000 | 47,702 | 2,907 | 58,015 | 25,558 |
| The rate of SSI of laparoscopic procedures, % | 4.33 | 2.03 | 8.29 | 7.77 | 11.5 |
| The rate of SSI of open procedures, % | 7.10 | 7.29 | 15.3 | 15.7 | 19.7 |
| The type of SSI in laparoscopic procedures in 2019, n, (%) (superficial/ deep/ organ, space) | 53/5/27 (62.4/5.9/ 31.8) | 71/3/16 (78.9/3.3/ 17.8) | 17/2/11 (56.7/6.7/ 36.7) | 299/25/134 (65.3/ 5.5/29.3) | 131/29/215 (34.9/ 7.7/57.3) |
| The type of SSI in open procedures in 2019, n, (%) (superficial/ deep/ organ, space) | 23/4/14 (56.1/9.8/ 34.1) | 49/4/24 (63.6/5.2/ 31.2) | 121/19/48 (64.4/ 10.1/25.5) | 424/79/172 (62.8/ 11.7/25.5) | 424/79/172 (62.8/ 11.7/25.5) |

All participating hospitals included all hospitals participating in each year

Core participating hospitals included all hospitals participating in all years from 2009 to 2019

Statistical analyses were performed using R version 4.0.3.

## Ethical considerations

Patient identifiers were de-identified in each hospital before data were submitted to JANIS. The anonymous data stored in the JANIS database were exported and analyzed, following approval from the Ministry of Health, Labour and Welfare (approval number 1007–6), according to Article 32 of the Statistics Act.

## Results

### The incidence of SSI among all participating hospitals

Fig 1 shows the average SSI rates for all surgeries, five digestive surgeries, and non-digestive surgeries (blue, orange, and green, respectively) from 2009 to 2019. The SSI rates for each year of surgery are listed in S1 Table. The average SSI rate for all surgeries among all participating hospitals for each year decreased significantly from 5.3% in 2009 to 3.5% in 2019 (p<0.001). The average SSI rate of the five digestive surgeries also decreased significantly from 11.8% to 8.1% (p<0.001), and the average SSI rate for the other surgeries decreased from 2.9% to 2.0% (p<0.001, Fig 1a).

### The incidence of SSI among core participating hospitals

In comparison, the average SSI rate of the core participating hospitals decreased significantly from 5.3% in 2009 to 3.3% in 2019 (p<0.0001, Fig 1b). The average SSI rate for the

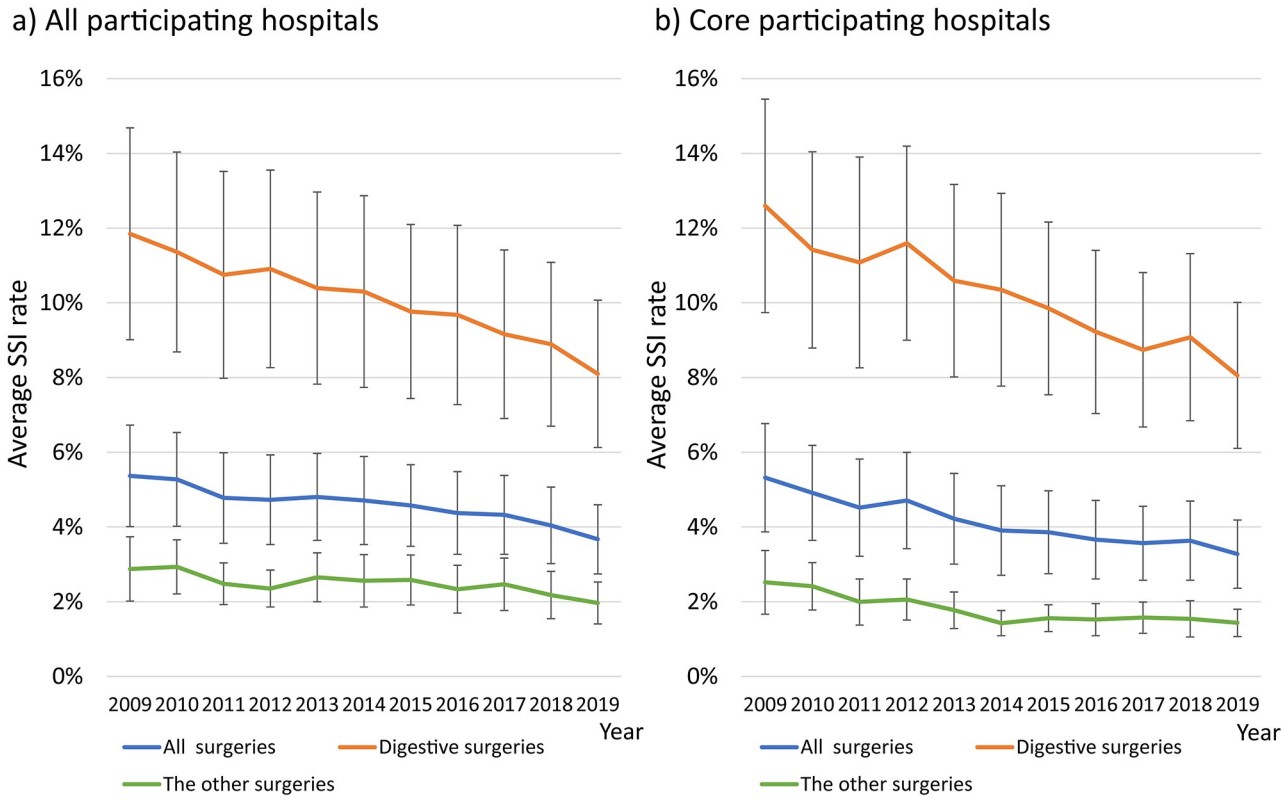

**Fig 1. Proportion of surgical site infection for the five digestive surgeries and other surgeries.** a) The proportion of surgical site infection in all participating hospitals. b) The proportion of surgical site infection in the core participating hospitals. The blue line shows the rate of surgical site infection for all types of surgeries. The orange line shows the rate of surgical site infection for five digestive surgeries (appendix surgery (APPY), gallbladder surgery (CHOL), colon surgery (COLO), small bowel surgery (SB), and rectal surgery (REC)). The green line shows the rate of surgical site infection for all except the five digestive surgeries. Error bars indicate standard deviation.

five digestive surgeries decreased significantly from 12.5% to 8.1% (p<0.0001), and the average SSI rate for all other surgeries decreased from 2.5% to 1.4% (p<0.0001). Overall, the results of the core hospitals were similar to those of all hospitals participating in the surveillance.

## Proportion of digestive surgeries using laparoscopic procedures

The proportion of laparoscopic procedures for each of the five digestive surgeries is shown in Fig 2. The proportion of laparoscopic procedures for CHOL in all participating hospitals was the highest, increasing from 69.1% in 2009 to 84.4% in 2019 (p<0.001, Fig 2a). The proportion of laparoscopic procedures of SB was the lowest, increasing from 4.85% to 20.1% (p<0.001, Fig 2a). The proportion of laparoscopic procedures performed for all other surgeries also significantly increased (p<0.0001, Fig 2a). Similar results were obtained for the core participating hospitals. The proportion of laparoscopic procedures for CHOL was the highest, increasing from 68.9% in 2009 to 82.3% in 2019 (p<0.0001, Fig 2b). The proportion of laparoscopic procedures for SB was the lowest: it increased from 5.38% to 21.4% (p<0.0001, Fig 2b). The proportion of laparoscopic procedures for REC in the core participating hospitals was only 3% higher than that in all participating hospitals (Fig 2a and 2b).

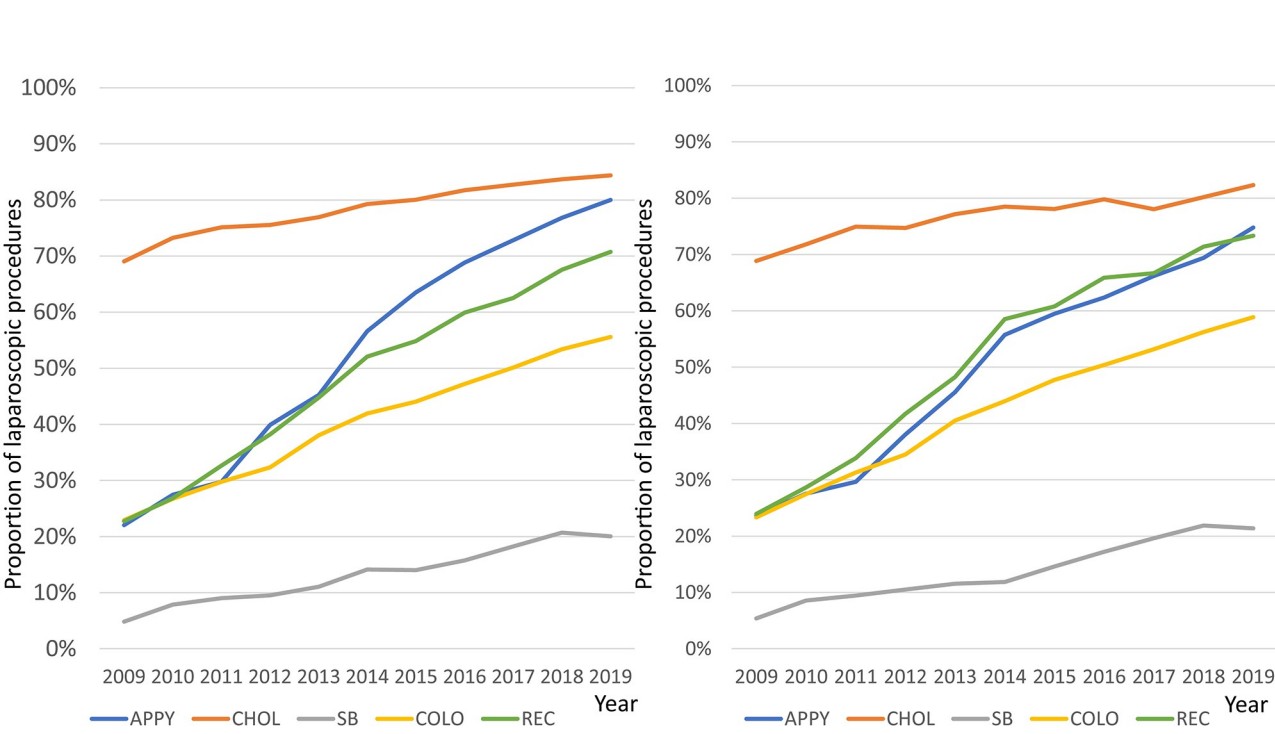

**Fig 2. Proportion of laparoscopic procedures for each of the five digestive surgeries.** a) Proportions of laparoscopic procedures for the five digestive surgeries in all participating hospitals. b) Proportions of laparoscopic procedures for the five digestive surgeries in the core participating hospitals The blue line indicates the rates for APPY. The orange line shows the rates of laparoscopic procedures performed for CHOL. The grey line shows the rates for SB. The yellow line shows the rates for COLO. The green line shows the rates for REC.

### Differences in the rates of SSI between laparoscopic and open procedures for each of the five digestive surgeries

The laparoscopic and open SSI rates for the five digestive surgeries for all and core participating hospitals are shown in Fig 3. The proportion of SSI for laparoscopic procedures for all and core participating hospitals was significantly lower than that for open procedures for the five digestive surgeries in all years from 2009 to 2019 (all five digestive surgeries: p<0.001). The average proportion of SSI for APPY for all participating hospitals was 4.3% for laparoscopic procedures from 2009 to 2019 and that for open procedures was 6.6%. The average proportion of SSI for CHOL for all participating hospitals was 1.9% for laparoscopic procedures, and that for open procedures was 7.5%. The average proportion of SSI for COLO for all participating hospitals was 7.5% for laparoscopic procedures, and that for open procedures was 15.7%. The average proportion of SSI for REC for all participating hospitals was 11.3% for laparoscopic procedures and 18.8% for open procedures. The average proportion of SSIs for SB for all participating hospitals was 7.9% for laparoscopic procedures, and that for open procedures was 15.2%. The proportion of SSIs for the core participating hospitals was almost the same as that for all participating hospitals (all five digestive surgeries: p<0.001).

### Contributions of laparoscopic procedures to the SSI rate

The odds ratios for the SSI rate when the proportion of laparoscopic procedures increased by 10%, calculated from the data of the core participating hospitals, is shown in Table 2. The odds

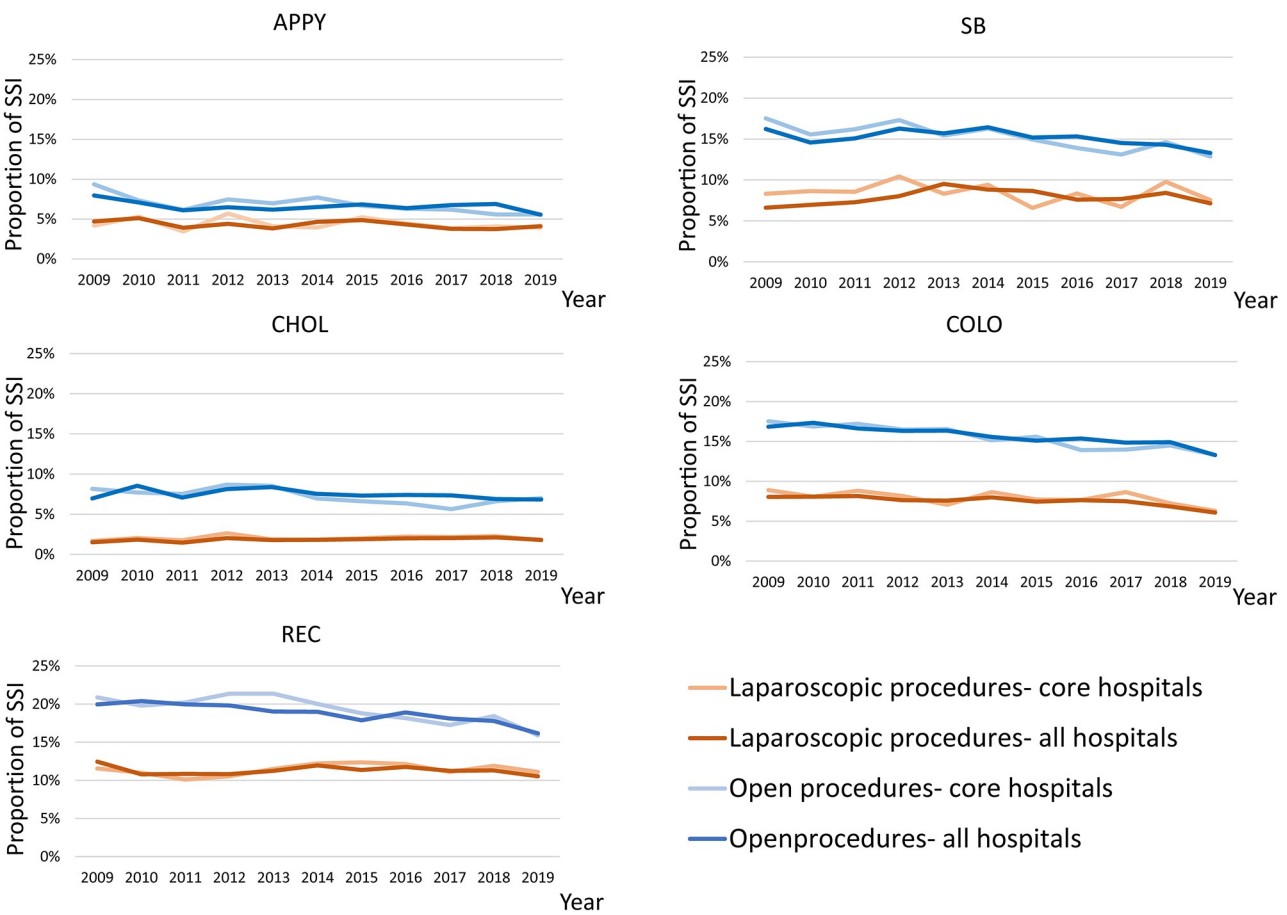

**Fig 3. Proportion of surgical site infection for each of the five digestive surgeries stratified by laparoscopic and other procedures, as well as the core and all hospitals.** The blue line shows the surgical site infection rate for open procedures in all participating hospitals. The light blue line shows the surgical site infection rate for open procedures in the core participating hospitals. The orange line shows the surgical site infection rate for laparoscopic procedures in all participating hospitals. The light orange line shows the surgical site infection rate for laparoscopic procedures in the core participating hospitals.

ratios for APPY, CHOL, SB, COLO, and REC were 0.948 (95% CI 0.931–0.965, p<0.001), 0.931 (95% CI 0.896–0.968, p<0.001), 0.934 (95% CI 0.901–0.969, p<0.001), 0.912 (95% CI 0.902–0.921, p<0.001), and 0.938 (95% CI 0.927–0.949, p<0.001), respectively. In each of the five digestive surgeries, an increase in the rate of laparoscopic procedures was significantly associated with a decrease in the SSI rate.

**Table 2. Odds ratio of the 10% increase in the proportion of laparoscopic procedures for the surgical site infection rates.**

| Surgical procedure | Odds Ratio | 95% Confidence Interval | | p-value |
|---|---|---|---|---|
| Appendix surgery | 0.948 | 0.931 | 0.965 | p<0.001 |
| Gallbladder surgery | 0.931 | 0.896 | 0.968 | p<0.001 |
| Small bowel surgery | 0.934 | 0.901 | 0.969 | p<0.001 |
| Colon surgery | 0.912 | 0.902 | 0.921 | p<0.001 |
| Rectal surgery | 0.938 | 0.927 | 0.949 | p<0.001 |

## Discussion

In this study, we investigated the impact of laparoscopic procedures in digestive surgeries on SSI reduction. We showed that the SSI rate continued to decline from 2009 to 2019 in all surgeries. The proportion of SSI was higher in the five digestive surgeries than in the others in each year, whereas the extent of the decrease in the SSI rate was larger in the five digestive surgeries than in the others. This study provided quantitative evidence that an increase in laparoscopic procedures is associated with a reduction in the SSI rate in digestive surgeries.

Our comparison of laparoscopic and open procedures over the 11-year period revealed that the SSI rate among laparoscopic procedures was significantly lower than that among open procedures, irrespective of the type of digestive surgery. The continuous increase in the proportion of laparoscopic procedures in each of the five digestive surgeries is likely to be a major factor underlying the decrease in the SSI rates for digestive surgeries. Laparoscopic procedures have been introduced because they are minimally invasive and are thus beneficial to patients. Kagawa et al. reported an increase in the number of laparoscopic procedures among colon surgeries from 2003 to 2015 [18]. However, prior to our study, there had been no report of annual trends in the proportions of laparoscopic procedures for the other four digestive surgeries.

Regarding REC, the percentage of superficial and deep incisional infections in laparoscopic procedures in 2019 was lower than that of open procedures (34.9, 7.7% vs 62.8, 11.7%). This showed that the SSIs were not related to the part of stoma. There was no difference between laparoscopic and open procedures in the percentage of type of SSIs of the other four surgeries.

Regarding SB, the proportion of laparoscopic procedures increased significantly from 5% in 2009 to 20% in 2019, but it was still much lower than that of the other four digestive surgeries. This likely reflects the difficulty in obtaining a definitive diagnosis in advance. On the contrary, the proportion of SSIs of upper gastrointestinal surgeries was variable (S1 Fig). Gastric surgeries were divided into total gastrectomy (GAST-T), Distal gastrectomy (GAST-D) and other gastric surgeries (GAST-O) in 2012. GAST-D and GAST-O showed a similar trend of increase in laparoscopic procedures associated with decrease in SSI rate, compared to the five digestive surgeries (odds ratio of the 10% increase in the proportion of laparoscopic procedures for the surgical site infection rates was 0.941 [95% CI: 0.914–0.968, p<0.001] and 0.947 [95%CI: 0.918–0.977, p<0.001], respectively). On the other hand, GAST-T and esophageal surgeries (ESOP) did not show the trend (p-value = 0.606 and 0.614). The number of procedures of GAST-T was continuously decreasing from 2,798 in 2012 to 1,570 in 2019. The number of laparoscopic procedures of GAST-T was also decreasing from 1,182 in 2012 to 704 in 2019. It was because of the development of the technique of gastrointestinal endoscopy, which led to detection and treatment of gastric diseases earlier. With respect to ESOP, the number of procedures of ESOP in 13 of 23 core participating hospitals was under 10 in 2019. Markar et al. reported that the outcome of esophageal surgeries in high surgical volume hospitals was better than that in low surgical volume hospitals, which suggests that surgeons' experience was required to reduce post-operative problems [19]. Meanwhile, regarding thoracic surgeries, although the proportion of thoracoscopic procedures increased similarly to that of laparoscopic procedures for digestive surgeries, the SSI rate among thoracoscopic procedures was not different from that of open procedures, probably because thoracic surgeries are usually clean and have a low SSI risk (S2 Fig). Similarly, although the frequency of endoscopic procedures increased in kidney and prostate surgeries, there was no difference in the SSI rates between endoscopic and open procedures.

We have shown the data of emergency and scheduled surgeries in 2019 in S2 Table. The proportion of laparoscopic procedures of emergency vs scheduled five digestive surgeries were 71.3% vs 84.0%, 76.7% vs 84.0%, 17.8% vs 25.1%, 23.9% vs 64.8%, and 28.2% vs 76.3%, for each

of the five procedures, respectively, which indicates that the proportion of laparoscopic procedures of emergency surgeries was significantly higher than that of scheduled surgeries. This result suggests that surgeons tend to choose open procedures in emergency situations. The rate of SSI of both laparoscopic and open procedures of emergency surgeries was higher than that of scheduled surgeries except for open procedures of APPY. Although laparoscopic procedures require surgical proficiency because of limited sight of instruments, they are known to reduce postoperative problems and promote early discharge from the hospital, and thus, should continue to increase in usage based on the cautious indications for each surgical procedure.

Our study suggests that the SSI rate can be reduced by performing laparoscopic procedures instead of open procedures for specific types of surgeries. Although it is unknown to what extent the proportion of laparoscopic procedures can be increased for each type of surgery, the increase observed from 2009 to 2019 could be used as a reference for the promotion of laparoscopic procedures.

This study has several limitations. First, the JANIS is a voluntary surveillance, which inevitably has selection biases, although approximately 10% of all hospitals in Japan were covered in the surveillance. Second, we could not collect data of patients' conditions, such as the type of antiseptic agent used or presence of bacterial colonization, because JANIS SSI section is designed to focus on SSIs. Third, the data collected in the surveillance have limitations regarding complexity of surgical procedures such as difficulty of the procedure, surgical technique used, and types of diseases that led to the surgery. Our study analysed a large dataset of 1,564,438 surgical procedures from 2009 to 2019, including 109,544 (APPY), 206,459 (CHOL), 60,225 (SB), 363,677 (COLO), and 134,695 (REC) procedures for the five digestive surgeries. This large-scale analysis based on the national surveillance database for over 10 years, for the first time, quantified the association between an increase in laparoscopic procedures and a reduction in the SSI rate in digestive surgeries. This report should encourage the appropriate introduction of laparoscopic surgical techniques in digestive surgeries and a consequent reduction of SSI rates.

## Supporting information

**S1 Fig. Proportion of thoracoscopic and laparoscopic procedures and surgical site infection rates in esophagial surgery and gastric surgery.** a) The proportion of thoracoscopic and laparoscopic procedures for esophageal (ESOP) and three types of gastric surgery (total gastrectomy: GAST-T, distal gastrectomy: GAST-D, other gastrectomy: GAST-O). The blue line shows the rate of ESOP in the core participating hospitals. The orange line shows the rate of GAST-T. The gray line shows GAST-D. The yellow line shows the rate of GAST-O. b-e) The blue line shows the surgical site infection rate for open procedures in all participating hospitals. The light blue line shows the surgical site infection rate for open procedures in the core participating hospitals. The orange line shows the surgical site infection rate for laparoscopic procedures in all participating hospitals. The light orange line shows the surgical site infection rate for laparoscopic procedures in the core participating hospitals.
(TIF)

**S2 Fig. Proportion of thoracoscopic procedures and surgical site infection rates in thoracic surgery.** a) The proportion of thoracoscopic procedures for thoracic surgery (THOR). The blue line shows the rate of thoracoscopic surgery in all participating hospitals and the light blue line shows the rate in the core participating hospitals. b) The blue line shows the surgical site infection rate for open procedures in all participating hospitals. The light blue line shows the surgical site infection rate for open procedures in the core participating hospitals. The

orange line shows the surgical site infection rate for thoracoscopic procedures in all participating hospitals. The light orange line shows the surgical site infection rate for thoracoscopic procedures in the core participating hospitals.
(TIF)

**S1 Table. The surgical site infection rates for each year for each surgery.**
(XLSX)

**S2 Table. The difference between emergency and scheduled surgeries.**
(XLSX)

## Acknowledgments

We are grateful to all the hospitals that participated and contributed data to JANIS; we are also grateful to Editage (www.editage.jp) for English language editing.

## Author Contributions

**Conceptualization:** Toshiki Kajihara.

**Data curation:** Toshiki Kajihara, Koji Yahara, Norikazu Kitamura.

**Formal analysis:** Toshiki Kajihara, Koji Yahara.

**Funding acquisition:** Motoyuki Sugai, Keigo Shibayama.

**Investigation:** Toshiki Kajihara, Koji Yahara.

**Methodology:** Toshiki Kajihara, Koji Yahara, Aki Hirabayashi.

**Project administration:** Toshiki Kajihara.

**Resources:** Toshiki Kajihara.

**Software:** Koji Yahara, Norikazu Kitamura.

**Supervision:** Koji Yahara, Yumiko Hosaka, Motoyuki Sugai, Keigo Shibayama.

**Validation:** Toshiki Kajihara.

**Visualization:** Toshiki Kajihara.

**Writing – original draft:** Toshiki Kajihara.

**Writing – review & editing:** Toshiki Kajihara, Koji Yahara, Aki Hirabayashi, Yumiko Hosaka, Norikazu Kitamura, Motoyuki Sugai, Keigo Shibayama.

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
