## [Decision Letter · Decision Letter 0]

19 Sep 2022

PONE-D-22-24040Association between the proportion of laparoscopic approaches for digestive surgeries and the incidence of consequent surgical site infections, 2009-2019: A retrospective observational study based on national surveillance data

PLOS ONE

Dear Dr. Toshiki Kajihara,

Thank you for submitting your manuscript to PLOS ONE. After careful consideration, we feel that it has merit but does not fully meet PLOS ONE’s publication criteria as it currently stands. Therefore, we invite you to submit a revised version of the manuscript that addresses the points raised during the review process.

We look forward to receiving your revised manuscript.

Kind regards,

Takehiko Hanaki, MD, PhD

Academic Editor

PLOS ONE

Journal Requirements:

"This study was supported by the Research Program on Emerging and Re-emerging Infectious Diseases from the Japan Agency for Medical Research and Development (AMED) under grant number JP21fk0108604. Funding acquisition: K. S. and M. S."

Additional Editor Comments:

Dear Dr. Toshiki Kajihara:

Thank you very much for allowing PLOS ONE team to see your submission entitled "Association between the proportion of laparoscopic approaches for digestive surgeries and the incidence of consequent surgical site infections, 2009-2019: A retrospective observational study based on national surveillance data".

We have now reviewed your submission and our view is that some major revisions are required before we can consider it for publication. This email is to invite you to respond to the reviewers comments attached below, and to revise your submission accordingly. We appreciate that these revisions will take some time and effort on your part, but we are confident that they will improve the quality and impact of your submission.

Reviewers' comments:

Reviewer's Responses to Questions

**Comments to the Author**

1. Is the manuscript technically sound, and do the data support the conclusions?

Reviewer #1: Partly

Reviewer #2: Yes

2. Has the statistical analysis been performed appropriately and rigorously? 

Reviewer #1: Yes

Reviewer #2: I Don't Know

3. Have the authors made all data underlying the findings in their manuscript fully available?

Reviewer #1: Yes

Reviewer #2: Yes

4. Is the manuscript presented in an intelligible fashion and written in standard English?

Reviewer #1: Yes

Reviewer #2: Yes

5. Review Comments to the Author

Reviewer #1: This is a large register-based study on surgical site infection (SSI) rate diagnosed in hospitals in Japan from 2009-2019. The study also report on the rate of laparoscopic procedures separately for this period. However, the study does not report the SSI rate after laparoscopic procedures. The study would benefit from a more detailed and transparent method section e.g. using RECORD statement, and maybe also other analyses depending on the data available.

Major comments:

1. It is not clear if your manuscript is reported in accordance with a reporting guideline, and the study design is not definitely clear to me. But I presume that RECCORD (https://www.equator-network.org/reporting-guidelines/record/) that is for register-based studies using routinely-collected health data would suit your manuscript and increase the transparency of your reporting.

2. Your outcome is somewhat unclear. You state that CDC definition was used, however, being unfamiliar with the JANIS SSI, it is unclear to me if the physicians at the hospital confirm SSI according to CDC when they report an SSI in JANIS SSI. Furthermore, it seems that JANIS SSI only include hospital data – what about the SSI treated at the patients private physician? Thus, I presume that you report the rate of SSI diagnosed in hospitals, correct?

3. Your study covers 10 years of data. It is evident from your data that the laparoscopic procedures increase tremendously during this period. However, could other factor contribute to decrease of SSIs? Patient-related factors, co-interventions such as prophylactic or empiric antibiotic treatment or others? There is no baseline information on the population receiving these procedures and the development in these over time. All your analyses are unadjusted. But this could be a limitation of your results. Please address in the result section or limitations section in the discussion depending on the data you have available.

4. If data are available you could address 3) by differentiating the types of procedures e.g. by using the surgical codes, if these are available.

Minor comments:

1. Introduction: Please ad a reference and be more specific than overall complications for the following sentence: “Laparoscopic techniques have been increasingly used in digestive surgeries, leading to fewer postoperative complications.”

2. Methods: Please add more information on the setting. You write “More hospitals progressively joined the JANIS SSI 91 section, and the program included 785 hospitals in 2019.” But what are the total number of hospitals in Japan, in other words how well is your coverage?

3. Methods: Are the procedures submitted to JANIS SSI reported by the hospital og linked by another nationwide register? How where they reported – as procedure codes? Which where used?

4. Methods: Who submit the outcome, SSI, according to CDC definitions? Hospitals, private practitioners? Is the outcome validated to follow CDC definitions?

5. Methods: How was a laparoscopic procedure confirmed?

6. Discussion: Please add a limitations section to your discussion.

Reviewer #2: The authors retrospectively investigated SSI ratio using their surveillance system form 2009 to 2019. I think several points should be addressed as follows.

How to collect all data from the hospitals participating in the surveillance？ EDC system? CRF by each year?

Please represent SSI data showing the differences of superficial, deep or organ.

How much was the SSI rate of upper GI surgery?

Rec shows high SSI rate, because of the stoma creation, or the incidence coming from the perineal wound?

The data of the difference between emergency and scheduled surgeries should be described.

What does the table 2 mean? Odd ratio was compared to that of the start year? The trend, fashion and equipment are changing year by year. It is difficult to compare the risk ratio.

The caption of the figure axis should be rewritten (e.g., "proportion" should be "Proportion" in Figure 2 and with high DPI figures).

6. PLOS authors have the option to publish the peer review history of their article (what does this mean?). If published, this will include your full peer review and any attached files.

Reviewer #1: **Yes: **Siv Fonnes

Reviewer #2: No

---

## [Author Response · Author response to Decision Letter 0]

21 Dec 2022

Point-by-point responses to the reviewers’ comments

Dear Editor and Reviewers:

Thank you for your efforts in reviewing our manuscript and for your helpful comments and suggestions. We have revised the manuscript in accordance with your comments. We hope that the changes, indicated in red font in the revised manuscript, reflect your suggestions and are satisfactory. Our point-by-point responses to your queries are given below.

Reviewer #1: This is a large register-based study on surgical site infection (SSI) rate diagnosed in hospitals in Japan from 2009-2019. The study also report on the rate of laparoscopic procedures separately for this period. However, the study does not report the SSI rate after laparoscopic procedures. The study would benefit from a more detailed and transparent method section e.g. using RECORD statement, and maybe also other analyses depending on the data available.

Major comments:

1. It is not clear if your manuscript is reported in accordance with a reporting guideline, and the study design is not definitely clear to me. But I presume that RECCORD (https://www.equator-network.org/reporting-guidelines/record/) that is for register-based studies using routinely-collected health data would suit your manuscript and increase the transparency of your reporting.

Response: We checked the RECORD guidelines that include 22 checklists. To satisfy the checklists, we have added the following three parts to the revised manuscript: 

(1) “in Japan” to the title for Item No. 1 in the checklist.

(2) “The SSI surveillance in JANIS is conducted using the definition of the US National Nosocomial Infections Surveillance System (NNIS), with some modifications. It is comparable to other international SSI surveillances, such as the OP-KISS of Germany [14] and the National Healthcare Safety Network of the United States of America [15]. Surgeons in hospitals performed procedures and diagnosed SSI cases and reported the data to JANIS according to NHSN Operative Procedure Categories using ICD-9-CM code (https://janis.mhlw.go.jp/section/master/surgicaltechniquecode_ver2.1_20121220.xls).” and “These included 188 of 408 (46.1%) hospitals with over 500 beds, 464 of 2,174 (21.3%) hospitals with 200–499 beds and 133 of 5790 (2.3%) hospitals with less than 200 beds.” to the Materials and Methods section for Item No. 6. (Page 6, Lines 97-104)

(3) “This study has several limitations. First, the JANIS is a voluntary surveillance, which inevitably has selection biases, although approximately 10% of all hospitals in Japan were covered in the surveillance. Second, we did not collect data of patients’ conditions, such as the type of antiseptic agent used or presence of bacterial colonization, because JANIS SSI section is designed to focus on SSIs. Third, the data collected in the surveillance have limitations regarding complexity of surgical procedures such as difficulty of the procedure, surgical technique used, and types of diseases that led to the surgery.” to the Discussion section for the Item No. 19. (Page 19, lines 311-317)

2. Your outcome is somewhat unclear. You state that CDC definition was used, however, being unfamiliar with the JANIS SSI, it is unclear to me if the physicians at the hospital confirm SSI according to CDC when they report an SSI in JANIS SSI. 

Response: Thank you for your comment. Regarding the definition, we have removed the statement of CDC definition, and instead added “The SSI surveillance in JANIS is conducted using the definition of the US National Nosocomial Infections Surveillance System (NNIS), with some modifications. It is comparable to other international SSI surveillances, such as the OP-KISS of Germany and the National Healthcare Safety Network of the United States of America.” to the Materials and Methods section. (Page 6, Lines 97-101) 

Most procedures of APPY, CHOL, SB, COLO, REC (defined by ICD-9-CM code) were performed in hospitals in Japan. Surgeons are usually responsible for the course of procedures in Japan. This is because JANIS collects the data from hospitals. The surgeons and/or infection control nurses in their hospitals validated the data. The surgeons at the hospital can confirm SSI because most patients are usually followed up at outpatient clinics in the hospitals in Japan. We have thus added “Most patients were followed up at outpatient clinics in their hospitals in Japan. The surgeon is usually responsible for the course of procedures in Japan. This is because JANIS collects the data from hospitals. The surgeons and/or infection control nurses in the hospitals validated the data and submitted the data every six months.” to the Materials and Methods section. (Page 7, Lines 108-111)

Furthermore, it seems that JANIS SSI only include hospital data – what about the SSI treated at the patients private physician? Thus, I presume that you report the rate of SSI diagnosed in hospitals, correct?

Response: Correct, they report the rate of SSI diagnosed in hospitals. The SSI treated by the patients’ private physician is not reported in JANIS SSI, although surgeons at hospitals can confirm SSI because most patients were usually followed up at outpatient clinics in the hospitals in Japan.

3. Your study covers 10 years of data. It is evident from your data that the laparoscopic procedures increase tremendously during this period. However, could other factor contribute to decrease of SSIs? Patient-related factors, co-interventions such as prophylactic or empiric antibiotic treatment or others? There is no baseline information on the population receiving these procedures and the development in these over time. All your analyses are unadjusted. But this could be a limitation of your results. Please address in the result section or limitations section in the discussion depending on the data you have available.

Response: Thank you for your comment. We could not collect patients’ disease stage, comorbidities, antibiotic use, etc., as JANIS was designed to monitor prevalence of nosocomial infections. We have added sentences regarding the limitation to the Discussion, as detailed in the response to your last comment below. 

4. If data are available you could address 3) by differentiating the types of procedures e.g. by using the surgical codes, if these are available.

Response: We use ICD-9-CM code for case definition. But we collected only NHSN Operative Procedure Categories, such as COLO, REC, etc. Therefore, we cannot analyze using the type of procedures for this study.

Minor comments:

1. Introduction: Please ad a reference and be more specific than overall complications for the following sentence: “Laparoscopic techniques have been increasingly used in digestive surgeries, leading to fewer postoperative complications.”

Response: We have added “In a systematic review, Buia et al showed that the advantage of the procedure was minimizing trauma to the abdominal wall compared with open surgery [5]. It was based on another systematic review, that pain intensity on VAS scale was reduced on day one and wound infections were decreased in laparoscopic procedures for suspected appendicitis [6]. Laparoscopic procedures accelerated the recovery after colon surgery by decreasing pain and duration of hospital stay [7]. Van der Pas et al. reported that patients in the laparoscopic colectomy group lost less blood, bowel function returned sooner, and hospital stay was shorter than those in the open surgery group in a randomised, phase 3 trial [8].” to the Introduction. (Pages 4-5, Lines 63-72)

2. Methods: Please add more information on the setting. You write “More hospitals progressively joined the JANIS SSI 91 section, and the program included 785 hospitals in 2019.” But what are the total number of hospitals in Japan, in other words how well is your coverage?

Response: We have added “These included 188 of 408 (46.1%) hospitals with over 500 beds, 464 of 2,174 (21.3%) hospitals with 200–499 beds and 133 of 5790 (2.3%) hospitals with less than 200 beds.” in the Materials and methods section. (Page 6-7, Lines 106-108)

3. Methods: Are the procedures submitted to JANIS SSI reported by the hospital og linked by another nationwide register? How where they reported – as procedure codes? Which where used?

Response: Yes, the procedures submitted to JANIS were reported by the hospital as procedure codes. We have added “Surgeons in hospitals performed procedures and diagnosed SSI cases and reported the data to JANIS according to NHSN Operative Procedure Categories using ICD-9-CM code (https://janis.mhlw.go.jp/section/master/surgicaltechniquecode_ver2.1_20121220.xls).” to the Materials and Methods section. (Page 6, Lines 101-104)

4. Methods: Who submit the outcome, SSI, according to CDC definitions? Hospitals, private practitioners? Is the outcome validated to follow CDC definitions?

Response: Surgeons in hospitals performed procedures and diagnosed SSI cases and submitted the data. We have added “Most patients were followed up at outpatient clinics in their hospitals in Japan. The surgeon is usually responsible for the course of procedures in Japan. This is because JANIS collect the data from hospitals. The surgeons and/or infection control nurses in their hospitals validated the data and submitted the data every six months.” to the Materials and Methods section. (Page 7, Lines 108-111)

5. Methods: How was a laparoscopic procedure confirmed?

Response: If laparoscopy was used in a patient’s procedure, a surgeon and/or an infection control nurse entered it in the dataset. We have added “The surgeons and/or infection control nurses in their hospitals validated the data and submitted the data every six months.” to the Materials and Methods section. (Page 7, Lines 110-111)

6. Discussion: Please add a limitations section to your discussion.

Response: Thank you for your comment. We have added “This study has several limitations. First, the JANIS is a voluntary surveillance, which inevitably has selection biases, although approximately 10% of all hospitals in Japan were covered in the surveillance. Second, we could not collect data of patients’ conditions, such as the type of antiseptic agent used or presence of bacterial colonization, because JANIS SSI section is designed to focus on SSIs. Third, the data collected in the surveillance have limitations regarding complexity of surgical procedures such as difficulty of the procedure, surgical technique used, and types of diseases that led to the surgery.” in the Discussion section. (Page 19, Lines 311-317)

Reviewer #2: The authors retrospectively investigated SSI ratio using their surveillance system form 2009 to 2019. I think several points should be addressed as follows.

How to collect all data from the hospitals participating in the surveillance？ EDC system? CRF by each year?

Response: We collected JANIS SSI division data from the hospitals using original EDC system every six months.

Please represent SSI data showing the differences of superficial, deep or organ.

Response: Thank you very much. We have added the type of SSI at the core hospitals in table 1.

How much was the SSI rate of upper GI surgery?

Response: We have added the data of upper GI surgeries in Table S1, and have added the following text “On the contrary, the proportion of SSIs of upper gastrointestinal surgeries was variable (Figure S1). Gastric surgeries were divided into total gastrectomy (GAST-T), Distal gastrectomy (GAST-D) and other gastric surgeries (GAST-O) in 2012. GAST-D and GAST-O showed a similar trend of increase in laparoscopic procedures associated with decrease in SSI rate, compared to the five digestive surgeries (odds ratio of the 10% increase in the proportion of laparoscopic procedures for the surgical site infection rates was 0.941 [95% CI: 0.914-0.968, p<0.001] and 0.947 [95%CI: 0.918-0.977, p<0.001], respectively). On the other hand, GAST-T and esophageal surgeries (ESOP) did not show the trend (p-value=0.606 and 0.614). The number of procedures of GAST-T was continuously decreasing from 2,798 in 2012 to 1,570 in 2019. The number of laparoscopic procedures of GAST-T was also decreasing from 1,182 in 2012 to 704 in 2019. It was because of the development of the technique of gastrointestinal endoscopy, which led to detection and treatment of gastric diseases earlier. With respect to ESOP, the number of procedures of ESOP in 13 of 23 core participating hospitals was under 10 in 2019. Marker SR et al. reported that the outcome of esophageal surgeries in high surgical volume hospitals was better than that in low surgical volume hospitals, which suggests that surgeons' experience was required to reduce post-operative problems.” in the discussion section. (Page 17, lines 270-286)

Rec shows high SSI rate, because of the stoma creation, or the incidence coming from the perineal wound?

Response: The high SSI rate in REC is not because of the stoma creation in both laparoscopic and open procedures according to the following data: among laparoscopic procedures, the percentage of superficial and deep incisional infections of rectal surgeries was less than that of organ and space infections (34.9% and 7.7%, compared with 57.3%, added to Table 1). Among open procedures, although the percentage of superficial incisional infections (62.8%) was higher than that of the others (11.7% for deep incisional infections and 25.5% for organ and space infections), breakdown of the superficial incisional infections into superficial Incisional Primary (SIP) and Secondary (SIS) confirmed that 124 of 131 superficial incision infections were SIP, indicating that most SSIs in REC procedures are not related to stoma creation. We have added “Regarding REC, the percentage of superficial and deep incisional infections in laparoscopic procedures in 2019 was lower than that of open procedures (34.9, 7.7% vs 62.8, 11.7%). This showed that the SSIs were not related to the part of stoma. There was no difference between laparoscopic and open procedures in the percentage of type of SSIs of the other four surgeries.” to the Discussion section. (Page 16, lines 262-266)

”

The data of the difference between emergency and scheduled surgeries should be described.

Response: Thank you for your comment. We have added Table S2 and the following sentences to the Discussion: “We have shown the data of emergency and scheduled surgeries in 2019 in Table S2. The proportion of laparoscopic procedures of emergency vs scheduled five digestive surgeries were 71.3% vs 84.0%, 76.7% vs 84.0%, 17.8% vs 25.1%, 23.9% vs 64.8%, and 28.2% vs 76.3%, for each of the five procedures, respectively, which indicates that the proportion of laparoscopic procedures of emergency surgeries was significantly higher than that of scheduled surgeries. This result suggests that surgeons tend to choose open procedures in emergency situations. The rate of SSI of both laparoscopic and open procedures of emergency surgeries was higher than that of scheduled surgeries except for open procedures of APPY.” (Pages 18, 294-302)

What does the table 2 mean? Odd ratio was compared to that of the start year? The trend, fashion and equipment are changing year by year. It is difficult to compare the risk ratio.

Response: The odds ratio was not compared to that of the start year. Rather, it estimates how the odds (p/(1-p) where p is probability of occurrence of SSI) changes when there is a 10% increase in the proportion of laparoscopic procedures, compared to that when there is no increase in it, across 11 years. 

The caption of the figure axis should be rewritten (e.g., "proportion" should be "Proportion" in Figure 2 and with high DPI figures).

Response: Thank you for your comment. We have corrected it to “Proportion” from “proportion”.

---

## [Decision Letter · Decision Letter 1]

1 Feb 2023

Association between the proportion of laparoscopic approaches for digestive surgeries and the incidence of consequent surgical site infections, 2009-2019: A retrospective observational study based on national surveillance data in Japan

PONE-D-22-24040R1

Dear Dr. Kajihara,

We’re pleased to inform you that your manuscript has been judged scientifically suitable for publication and will be formally accepted for publication once it meets all outstanding technical requirements.

Kind regards,

Takehiko Hanaki, MD, PhD

Academic Editor

PLOS ONE

Additional Editor Comments (optional):

Reviewers' comments:

Reviewer's Responses to Questions

**Comments to the Author**

1. If the authors have adequately addressed your comments raised in a previous round of review and you feel that this manuscript is now acceptable for publication, you may indicate that here to bypass the “Comments to the Author” section, enter your conflict of interest statement in the “Confidential to Editor” section, and submit your "Accept" recommendation.

Reviewer #1: All comments have been addressed

Reviewer #2: All comments have been addressed

2. Is the manuscript technically sound, and do the data support the conclusions?

Reviewer #1: Yes

Reviewer #2: Yes

3. Has the statistical analysis been performed appropriately and rigorously? 

Reviewer #1: Yes

Reviewer #2: Yes

4. Have the authors made all data underlying the findings in their manuscript fully available?

Reviewer #1: Yes

Reviewer #2: Yes

5. Is the manuscript presented in an intelligible fashion and written in standard English?

Reviewer #1: Yes

Reviewer #2: Yes

6. Review Comments to the Author

Reviewer #1: Thank you for addressing all comments and for writing a more transparent method section and for adding a limitation section in the discussion section.

Reviewer #2: (No Response)

7. PLOS authors have the option to publish the peer review history of their article (what does this mean?). If published, this will include your full peer review and any attached files.

Reviewer #1: **Yes: **Siv Fonnes

Reviewer #2: No

---

## [Editor Report · Acceptance letter]

9 Feb 2023

PONE-D-22-24040R1 

Association between the proportion of laparoscopic approaches for digestive surgeries and the incidence of consequent surgical site infections, 2009-2019: A retrospective observational study based on national surveillance data in Japan 

Dear Dr. Kajihara:

I'm pleased to inform you that your manuscript has been deemed suitable for publication in PLOS ONE. Congratulations! Your manuscript is now with our production department. 

Kind regards, 

on behalf of

Dr. Takehiko Hanaki 

Academic Editor

PLOS ONE